# Endoscopic and Histological Characteristics of Gastric Cancer Detected Long After *Helicobacter pylori* Eradication Therapy

**DOI:** 10.3390/cancers16244153

**Published:** 2024-12-13

**Authors:** Ryo Abe, Shu Uchikoshi, Yohei Horikawa, Nobuya Mimori, Yuhei Kato, Yuta Tahata, Saki Fushimi, Masahiro Saito, Satsuki Takahashi

**Affiliations:** 1Department of Gastroenterology, Hiraka General Hospital, Yokote 013-8610, Akita, Japan; yakyu2727@yahoo.co.jp (R.A.); riderboo@violet.plala.or.jp (N.M.); burst_error3rd@s6.dion.ne.jp (Y.K.); m05059ytakt@gmail.com (Y.T.); saki3040@gmail.com (S.F.); 2Department of Gastroenterology, Graduate School of Medicine, Akita University, Akita 010-8543, Akita, Japan; 3Matsuzono Second Hospital, Morioka 020-0103, Iwate, Japan; shu_progressing@me.com; 4Crea Clinic, Sendai 980-0021, Miyagi, Japan; 5Department of Clinicopathology, Hiraka General Hospital, Yokote 013-8610, Akita, Japan; hrkpatho@rnac.ne.jp (M.S.); sasasastsk@pat.hi-ho.ne.jp (S.T.)

**Keywords:** pE-GCs, late cancer, the remnant rate of the fundic glands

## Abstract

The risk of post-eradication gastric cancers (pE-GCs) has increased. The annual incidence rate of pE-GCs is 0.24%. pE-GCs include cancers that develop immediately and long after *Hp* eradication therapy. Although late cancers have a particular nature as pE-GCs, the details of their features remain unclear. Therefore, we aimed to clarify the endoscopic and histological characteristics of late pE-GCs. Comparing an immediate and a delayed group, there was no significant difference regarding the background and tumor mucosa. In the delayed group, the remnant rate of the fundic glands was 19.8 ± 15.6%, IM was 87.1%, and 90.3% of the patients exhibited persistent neutrophil infiltration. This study suggested that patients with poor improvement in the fundic gland and IM and persistent inflammation are at high risk for developing pE-GCs.

## 1. Introduction

Based on previous studies, the indication for *Helicobacter pylori* (*Hp*) eradication therapy was once limited to patients with conditions such as peptic ulcers, gastric cancers after endoscopic resection, idiopathic thrombocytopenic purpura, and mucosa-associated lymphoid tissue lymphoma. However, considering its preventive effects against metachronous gastric cancers [1,2], eradication therapy for *Hp*-associated gastritis has been covered by the National Health Insurance of Japan since 2013. Since then, the utilization rate of *Hp* eradication therapy (*Hp*-ET) has significantly increased, and it reached 1.5 million in 2016 [3]. As a result, the risk of post-eradication gastric cancers (pE-GCs) has increased. The annual incidence rate of gastric cancer after successful eradication therapy is 0.24% [4].

In general, pE-GCs have several similarities with *Hp*-positive gastric cancer. Thus, pE-GCs arises from *Hp*-infected gastric mucosa that was previously infected with *Hp* [5]. It has been reported that the endoscopic characteristics of pE-GCs (1) are common in the antrum of stomach, (2) are of the depressed type, (3) are small lesions with a tumor diameter of <20 mm, (4) have unclear demarcation, and (5) are of the differentiated type with a gastric phenotype [6]. However, they also have other endoscopic and histopathologic characteristics that are different from those of *Hp*-positive gastric cancer [7,8]. The specific endoscopic feature of pE-GCs was a gastritis-like appearance, which is characterized by uniform papillae and/or tubular pits with a whitish border, regular or faint micro vessels, and unclear demarcation [7]. The specific histological feature of pE-GCs is an epithelium with low-grade atypia, which often appears on the surface of them, where gastric-type mucin is frequently expressed [9]. Based on these features, the *Hp*-infected gastric mucosa is secondarily modified histologically by *Hp*-ET [10].

Otherwise, pE-GCs include cancers that develop immediately and long after *Hp* eradication therapy [11,12]. Most pE-GCs that develop immediately after *Hp* eradication therapy are already present during treatment, and it is reasonable to assume that they developed after *Hp*-ET. Since immediate cancer can include both these cancers and cancers that have developed purely after eradication, late cancers have been thought to have a particular nature as pE-GCs. However, the details of the features of late cancers remain unclear and there are few studies on the characteristics of the background mucosa of late cancers [13]. In this study, we focused on the remnant rate of the fundic glands and intestinal metaplasia (IM), crypt enlargement, and neutrophil infiltration of the background mucosa.

As mentioned above, the number of patients who develop cancer after *Helicobacter pylori* eradication is increasing, and many of them are currently endoscopically followed up once every few years for the purpose of screening for pE-GCs. It is clinically important to identify the specific features of pE-GCs and their background mucosa for risk stratification and effective surveillance. Therefore, we hypothesized that late pE-GC has a specific nature that is different from that of immediate cancer. Further, this study aimed to evaluate the endoscopic and histological characteristics of the two types of pE-GC and their background.

## 2. Methods

### 2.1. Patients

The endoscopic images and histological specimens of patients with differentiated gastric cancers detected after *Hp*-ET who underwent gastric endoscopic submucosal dissection (ESD) at Hiraka General Hospital between January 2015 and December 2023 were compared. Data were collected from the hospital database and retrospectively reviewed. Figure 1 shows the annual transitions of pE-GCs treated via gastric ESD, with a total of 100 cases. In cases of multiple gastric cancers, we investigated only one representative object (e.g., the largest lesion or the deepest lesion). The incidence of pE-GC was bimodal 7 years after *Hp*-ET. pE-GCs were abundant by 1 year after eradication, and remained the same thereafter, but decreased drastically after 7–8 years, and began to increase again after 9 years. Figure 2 shows a flow diagram of this research. Patients with an unknown detail on eradication history (n = 12) and those with diffuse-type gastric cancer (n = 8) were excluded from the analysis. The remaining patients were divided into the immediate group (n = 69), which included those with cancer detected within 6 years after *Hp*-ET, and the delayed group (n = 31), which included those with cancer detected >6 years after *Hp*-ET. The degree of endoscopic gastric mucosal atrophy was assessed using the Kimura and Takemoto classification [14].

Tumors were categorized according to location (i.e., upper [U], middle [M], or lower third [L] of the stomach), and circumference (i.e., anterior wall [A], posterior wall [P], greater curvature [G], or lesser curvature [L] of the stomach). Measurement and histological classification were performed on the resected specimens according to the criteria of the Japanese Gastric Cancer Association [15]. Curability was evaluated according to the Japanese Gastric Cancer Treatment Guidelines, 5th Edition [16]. *Hp* infection was assessed in all patients using at least one of the following three methods: (1) the anti-*Hp* immunoglobulin G serological test, (2) the rapid urease test, or (3) the ^13^C-urea breath test.

### 2.2. Evaluation of Endoscopic Findings and Histological Features

The background mucosa and tumor mucosa of the ESD specimens were examined individually based on the endoscopic and histological findings. The background mucosa was defined as the edge of the ESD specimens that does not include tumor tissues. The tumor mucosa was defined as the center of the lesion. The endoscopic images of the same portion of the above-defined specimens were reviewed and assessed.

The endoscopic findings of the background mucosa were as follows: enlarged fold, map-like redness, intermediate zone irregularity, and the presence of a regular arrangement of collecting venules and a light blue crest [17,18]. The endoscopic findings of the tumor mucosa were as follows: an irregular surface structure, an irregular vascular and surface pattern, and a gastritis-like appearance [7,19].

The histological findings of the background mucosa were as follows: a low remnant rate of the fundic glands, intestinal metaplasia, crypt enlargement, and neutrophil infiltration [20]. The histological findings of the tumor mucosa were as follows: mosaicism, elongation of the noncancer ducts, and an overlying non-neoplastic epithelium [8,21]. The Remnant rate of the fundic glands is the original criterion used as an indicator of histological gastric atrophy. It was calculated based on the microscopically measured area of the fundic glands on the edge of the ESD specimen (length: 1.0 mm × width: 2.0 mm, ×40), using Photoshop ver.4.0.1 (Figure 3).

All the endoscopic examinations were performed by four expert endoscopists certified by the Japanese Society of Gastroenterology (YT, NM, SF, and YH). The endoscopic findings were evaluated by each of the endoscopists independently on basis of the endoscopic images. Histological evaluation was performed by expert pathologists (MS and ST) independently of the endoscopists. When there was no agreement, a final decision was obtained via a consensus after a discussion of each individual case. During the evaluation of the endoscopic images, the endoscopists and pathologists were blinded to any of the background characteristics.

### 2.3. Statistical Analysis

Statistical analysis of the clinical and endoscopic data was performed using the chi-square test for categorical data and the Student’s t-test for numerical data in the univariate analysis. The absolute differences and *p*-values were examined. A *p* value of < 0.05 was considered statistically significant. All statistical analyses were performed with JMP version 12.0 (SAS Institute, Cary, NC, USA).

### 2.4. Ethics Approval

The current study was performed in accordance with the principles of the Declaration of Helsinki. The ethics committee of the Hiraka General Hospital approved this study (no. 2062022). Optout was used to assess and publish the data.

## 3. Results

Table 1 shows the baseline characteristics of the immediate and delayed groups. The delayed group (72 [52–80]) was older than the immediate group (68 [39–80]) (*p* = 0.014). In addition, the lesions in the delayed group (48.4%, 15/31) were likely to present with lesions in the middle part of the stomach more frequently than those in the immediate group (29.0%, 20/69) (*p* = 0.004). The delayed group (54.8%, 14/31) had more experience of endoscopic resection (ER) than the immediate group (8.7%, 6/69) (*p* = 0.000). Patients in the delayed group (9.7%, 3/31) had more multiple gastric cancers at the same time than those in the immediate group (5.8%, 4/69) (*p* = 0.018). There were no significant differences in terms of sex distribution, previous history of smoking, drinking, peptic ulcers, acid secretion inhibitors, endoscopic gastric atrophy, lesion circumference, macroscopic type, specimen size, and curability (eCura) between the immediate and delayed groups. In particular, the two groups had a high rate of endoscopic extended gastric atrophy (O-II, O-III), which is known as a precancerous status (73.9% vs. 67.7%).

There was no significant difference in terms of the endoscopic and histological findings of the background mucosa between the immediate and delayed groups (Table 2). Notably, based on the background mucosa, in the delayed group, the remnant rate of the fundic glands was extremely low (19.8 ± 15.6%), and IM was still observed in 87.1% (27/31) of the patients. Further, 90.3% (28/31) of the patients presented with persistent neutrophil infiltration long after *Hp* eradication therapy (Table 2). Additionally, in both groups, most of the IM cases were associated with neutrophil infiltration (79.4% in the immediate group, 74.1% in the delayed group) (Table 2). Similarly to those of the background mucosa, the endoscopic and histological findings of the tumor mucosa did not significantly differ between the two groups (Table 3).

Figure 4 depicts the annual transitions of the remnant rate of the fundic glands based on the background mucosa. During the observation years, the remnant rate of the fundic glands based on the background mucosa did not improve even long after *Hp* eradication therapy. pE-GCs occurred in the atrophic mucosa with a low remnant rate of the fundic glands.

## 4. Discussion

Initially, we assumed that pE-GCs that develop long after *Hp*-ET have characteristic findings that differ from those of pE-GCs that occur immediately after *Hp*-ET. However, similar characteristics of the tumors occurred on same characteristics of the background mucosa over a long period of time after eradication therapy in the present study. Based on this result, late cancer might develop on background mucosa where inflammation persisted, even after *Hp*-ET, without improvement of fundic gland atrophy and IM over the years. To the best of our knowledge, this is the first report showing that the remnant rate of the fundic glands in the gastric mucosa where pE-GCs arise does not improve over time. Several reports have used biopsy specimens for the histological evaluation of atrophy. However, this report used ESD specimens, which included the entire mucosal layer and the submucosal layer up to just above the muscle layer. Therefore, they facilitated the detailed examination of the whole mucosa over a wide area.

In relation to concerns regarding the tumor mucosa, previous reports showed that the notable endoscopic features of pE-GCs were a smaller size, a depressed appearance, a differentiated type, and a location in the ML area [12,21,22]. The tumor characteristics in our study were similar to those reported in previous studies. Kobayashi et al. [7] revealed that 44% of patients with pE-GCs presented with a gastritis-like appearance. In our study, 65.2% of patients in the immediate group and 67.7% in the delayed group exhibited a gastritis-like appearance, which is a characteristic endoscopic finding of pE-GCs. Kitamura et al. [9] revealed that 81% of patients with pE-GCs had an epithelium with low-grade atypia. Further, in this study, 39.1% of patients in the immediate group and 41.9% in the delayed group presented with a histological overlying non-neoplastic epithelium, which was slightly lower than in a previous report.

Gastric atrophy and IM induced by chronic *Hp* infection are recognized as risk factors for pE-GCs [12,23,24,25]. *Hp*-ET could improve gastric mucosal atrophy and reduce the risk of pE-GC development if the condition is not severe [25,26]. However, there was no improvement in advanced atrophy after *Hp*-ET, and advanced atrophy could be a risk factor for pE-GCs [25,26,27]. The current study aimed to investigate histological atrophy using the remnant rate of the fundic glands. Based on the endoscopic findings, the immediate and delayed groups presented with moderate to severe atrophy of the mucosa. According to the histological findings, the remnant rate of the fundic glands did not improve. Toyokawa et al. [28] reported that atrophic gastritis can improve after approximately 9 years. Thus, patients can be at high risk of pE-GCs for quite some time from the perspective of atrophy being a risk factor.

IM on the lesser curvature of the gastric corpus was associated with a higher risk of developing pE-GCs [29,30]. In several cases, there was no significant improvement in IM long after *Hp* eradication therapy [31,32,33]. Gastric IM was associated with an impaired para-cellular barrier, and irreversible mucosal barrier dysfunction may cause sustained trans-epithelial penetration of various intra-gastric substances, which may continue to present as inflammation [34,35]. In this study, pE-GCs with IM were significantly high not only in the immediate group but also in the delayed group, and the delayed group had persistent neutrophil infiltration, which should be controlled by *Hp*-ET. For these reasons, it is suggested that inflammation of the gastric mucosa persists on IM that is not improved by *Hp* eradication therapy and might be a risk factor for the carcinogenesis of pE-GCs.

Cancer occurrence after *Hp* eradication therapy has been reported to vary from several months to more than 10 years, and the risk of gastric cancer did not change in the second decade of follow-up for intestinal-type gastric cancer regardless of the grade of baseline gastric mucosal atrophy [12,26]. However, there are few studies on the characteristics of detailed features and the background mucosa of late cancers. Kamada T et al. [4] revealed that pE-GCs detected within 5 years were likely to develop into differentiated cancer on the background mucosa with advanced atrophy, similar to *Hp*-positive gastric cancer. Haruma K et al. [13] reported in a Japanese study that pE-GCs which are found more than 10 years later occur in men who have atrophic gastritis in the gastric body before eradication, and are differentiated cancers in the non-cardia region. In our study, pE-GCs of the delayed group were likely to present with lesions in the middle third of the stomach more frequently than those in the immediate group, with a significant difference, and a lesion in the upper third of the stomach only occurred in one case (3.3% (1/31)). These results support previous reports. In addition, the delayed group had more experience of ER and had more multiple gastric cancers at the same time than the immediate group in this study, although there are no previous reports to support these results. From the above, we should pay more attention to lesions in the middle third of the stomach, post-ER cases, and multiple lesions when monitoring pE-GCs during surveillance endoscopy.

The current study had several limitations. First, it was performed in a single institution, and only a relatively small number of patients were retrospectively enrolled. Second, this report used ESD specimens for the assessment of background mucosa. The findings on background mucosa might change depending on the site of the lesion. Third, this study did not examine mucus characteristics and genetic/epigenetic alteration. The background tissue had genetic/epigenetic alterations, which is currently considered a reliable mechanism of gastric carcinogenesis [36]. In the future, it is desirable to investigate the effectiveness of risk stratification by other methods using background mucosal methylation, gene mutations, and gene expression profiling. Finally, only patients with pE-GCs were examined, and these patients were not compared with those without pE-GCs. In the future, a multicenter large-scale prospective study about the association between existing risks and genetic/epigenetic alterations should be performed to determine the risk stratified for a large number of patients who have received *Hp*-ET.

## 5. Conclusions

In addition to previous reports, this study showed that patients with a low remnant rate of the fundic glands and IM and persistent mucosal inflammation in the mucosa were at high risk for developing gastric cancer after *Hp*-ET. In addition to research on endoscopic/histological high-risk groups, integrating genetic/epigenetic alteration studies, such as DNA methylation of the background mucosa, can help further identify high-risk groups.

## Figures and Tables

**Figure 1 cancers-16-04153-f001:**
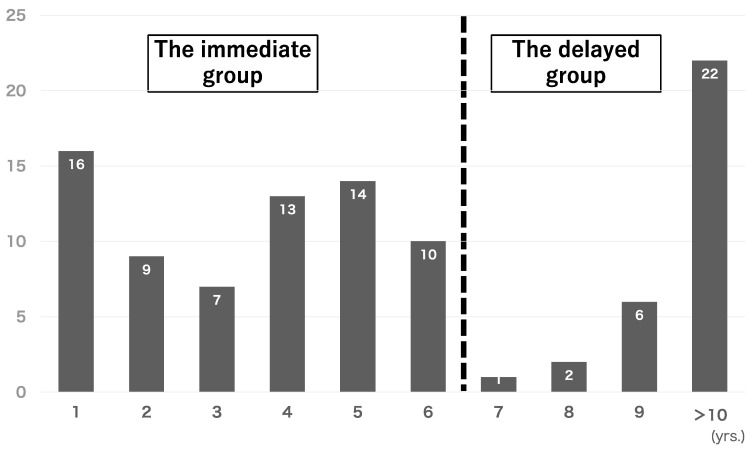
Annual transitions of post-eradication gastric cancer treated with gastric ESD (total 100 cases).

**Figure 2 cancers-16-04153-f002:**
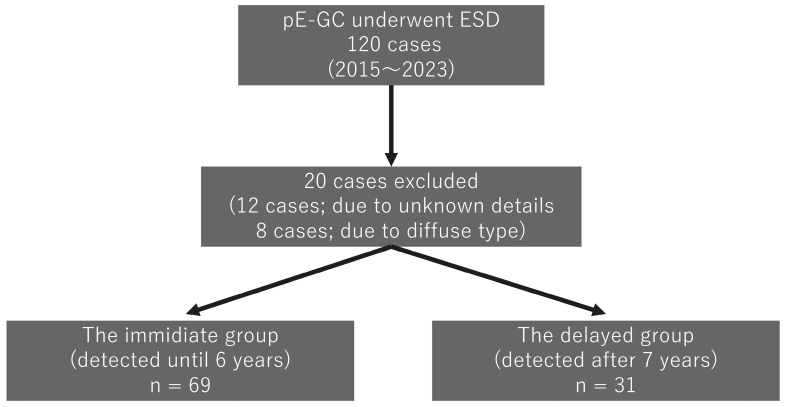
Flow diagram of patient enrollment.

**Figure 3 cancers-16-04153-f003:**
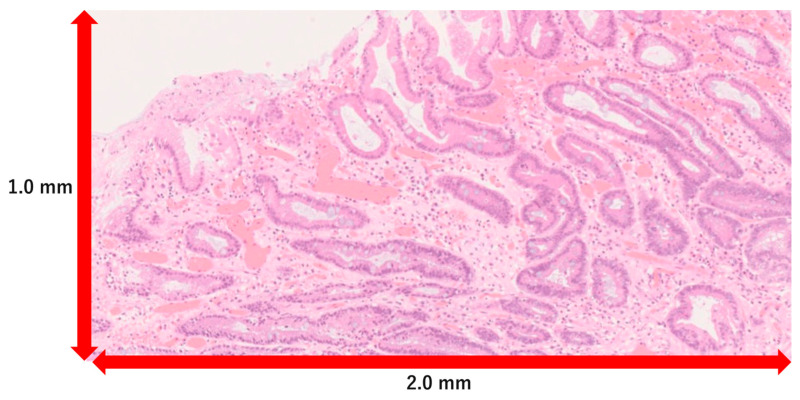
The remnant rate of the fundic glands. This was calculated by microscopically measuring the area of the fundic glands on the edge of the ESD specimen (1.0 mm × 2.0 mm).

**Figure 4 cancers-16-04153-f004:**
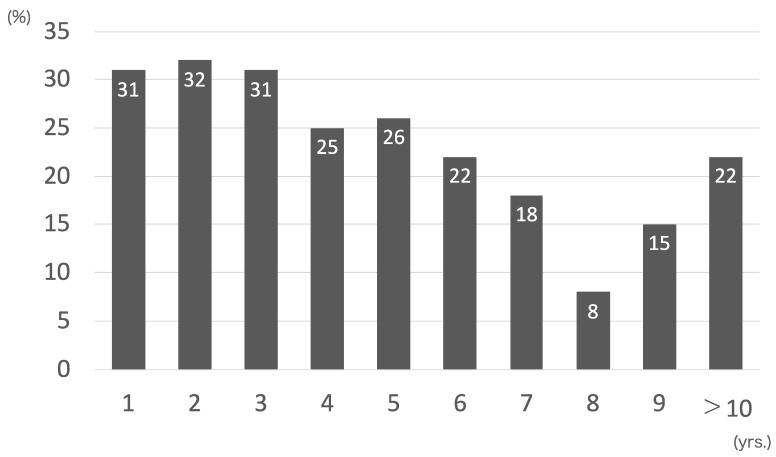
Annual transitions of remnant rate of fundus glands on background mucosa.

**Table 1 cancers-16-04153-t001:** Baseline characteristics of patients with post-eradication gastric cancer who underwent ESD.

	The Immediate Group	The Delayed Group	*p*
	(n = 69)	(n = 31)	
**Patient characteristics**			
Age, median [range] (y)	68 [39–80]	72 [52–80]	0.014 *
Sex			
Male/Female	53/16	23/8	0.745 ^†^
Previous history (positive/negative)			
Smoking	33/36	20/11	0.071 ^†^
Drinking	44/25	25/6	0.136 ^†^
Peptic ulcers	4/65	3/18	0.351 ^†^
Acid secretion inhibitor	30/39	17/14	0.249 ^†^
Endocopic resection	6/63	14/17	0.0004 ^†^
Multiple gastric cancers	4/65	3/28	0.018 ^†^
Endoscopic gastric atrophy			
C-I, II/C-III, O-I/O-II, III	1/17/51	1/9/21	0.541 ^†^
**Lesion characteristics**			
Lesion location			
U/M/L	8/20/41	1/15/15	0.004 ^†^
Lesion circumference			
A/P/G/L	9/14/16/30	3/3/5/20	0.337 ^†^
Macroscopic type			
Elevated/Flat/Depressed	13/5/51	7/3/21	0.937 ^†^
Specimen size			
mean [SD] (mm^2^)	250 [435]	200 [772]	0.080 *
Curability (eCura)			
A/B/C1/C2	61/5/0/3	24/3/0/4	0.813 ^†^

U, upper third; M, middle third; L, lower third of the stomach; A, anterior wall; P, posterior wall; G, greater curvature; L, lower curvature; SD, standard deviation; *, Student’s *t*-test; ^†^, Chi-square test.

**Table 2 cancers-16-04153-t002:** Endoscopic and histological findings of background mucosa.

Background Mucosa		The Immediate Group	The Delayed Group	*p*
		(n = 69)	(n = 31)	
Endoscopic findings	enlarged fold	31 (44.9%)	13 (41.9%)	0.54 *
	map-like redness	31 (44.9%)	13 (41.9%)	0.54 *
	intermediate zone irregularity	17 (24.6%)	10 (32.3%)	0.31 *
	regular arrangement of collecting venules	0 (0.0%)	0 (0.0%)	1.00 *
	light blue crest	61 (88.4%)	24 (77.9%)	0.19 *
Histological findings	the remnant rate of the fundic glands [SD]	26.1 [22.2]%	19.8 [15.6]%	0.62 *
	crypt enlargement	46 (63.7%)	25 (80.6%)	0.19 *
	intestinal metaplasia	63 (91.3%)	27 (87.1%)	0.23 *
	neutrophil infiltration	51 (73.9%)	28 (90.3%)	0.10 *
	intestinal metaplasia with neutrophil infiltration	50 (79.4%)	20 (74.1%)	0.245 *

SD, standard deviation; *, Student’s *t*-test.

**Table 3 cancers-16-04153-t003:** Endoscopic and histological findings of tumor mucosa.

Tumor Mucosa		The Immediate Group	The Delayed Group	*p*
		(n = 69)	(n = 31)	
Endoscopic findings	surface structure (villous/pit/mixed)	22:15:32	8:7:16	0.38 ^†^
	irregular vascular pattern	56 (81.2%)	19 (61.3%)	0.07 *
	irregular surface pattern	8 (11.6%)	4 (12.9%)	0.55 *
	a gastritis-like appearance	45 (65.2%)	21 (67.7%)	0.42 *
Histological findings	mosaicism	37 (53.6%)	20 (64.5%)	0.21 *
	elongation of the noncancer ducts	0 (0.0%)	0 (0.0%)	1.00 *
	overlying non-neoplastic epithelium	27 (39.1%)	13 (41.9%)	0.48 *

*, Student’s *t*-test; ^†^, Chi-square test.

## Data Availability

The data presented in this study are available in this article.

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
