# Peer review of "Endoscopic and Histological Characteristics of Gastric Cancer Detected Long After Helicobacter pylori Eradication Therapy"

_cancers, 2024, doi:10.3390/cancers16244153_

Round 1
Reviewer 1 Report
Comments and Suggestions for Authors
The authors divided endoscopically resected eradicated gastric cancer into the immediate group and the delayed group, and examined the endoscopic and histological characteristics of each group. In the delayed group, the remnant rate of the fundic gland was low, and intestinal metaplasia (IM) and persistent high-frequency neutrophil infiltration were observed. They state that these may be risk factors for post-eradication gastric cancer.
Comments
1. In the analysis of posteradication gastric cancer, the patient background, including endoscopic resection history of gastric cancer (including adenoma), history of surgical resection of gastric cancer, history of gastric and duodenal ulcer, history of other cancers, family history of gastric cancer, smoking and drinking history, and acid secretion inhibitor (for example, in the last 3 months or more) is important. Please explain and present these informations as much as possible. Also, are there any cases of multiple gastric cancers in this study?
2. The authors state that the poor improvement of fundic gland atrophy and IM is a risk factor for late cancer. However, the authors did not check the state of the gastric mucosa before the eradication, so please revise the description.
3. Please describe the reason for dividing the cases into the immediate group and the delayed group based on whether they were eradicated within 6 years or after more than 7 years.
4. Please explain the reason for excluding diffuse-type gastric cancer.
5. The authors state the presence of IM as a mechanism that causes neutrophil infiltration of the gastric mucosa, but is there a link between neutrophil infiltration and the presence of IM in this study?
6. As a limitation of this study, please include that the background tissue was analyzed using ESD resection specimens.
7. Please check whether the asterisks in Line 120 and 122 are necessary.
Author Response
Comments
- In the analysis of posteradication gastric cancer, the patient background, including endoscopic resection history of gastric cancer (including adenoma), history of surgical resection of gastric cancer, history of gastric and duodenal ulcer, history of other cancers, family history of gastric cancer, smoking and drinking history, and acid secretion inhibitor (for example, in the last 3 months or more) is important. Please explain and present these informations as much as possible. Also, are there any cases of multiple gastric cancers in this study?
Thank you for pointing this out. We agree with this comment. Therefore, according to reviewer’s comment, we revised Table1 and Result part of manuscript. History of smoking, Drinking, Peptic ulcers, acid secretion inhibitor and endoscopic resection were added to Table1. Also, multiple gastric cancers at the same time was added to Table1. Further, descriptions as to Table1 were revised as follows: The delayed group (54.8%, 14/31) had more experience of endoscopic resection than the immediate group (8.7%, 6/69) (p = 0.000). Patients of the delayed group (9.7%, 3/31) had more multiple gastric cancers at the same time than that of the immediate group (5.8%, 4/69) (p = 0.018). There were no significant differences in terms of sex distribution, previous history of smoking, drinking, peptic ulcer, acid secretion inhibitor, endoscopic gastric atrophy, lesion circumference, macroscopic type, specimen size and Curability (eCura) between the immediate and delayed group. Unfortunately, history of other cancers and familial history of gastric cancers were unavailable for our database.
- The authors state that the poor improvement of fundic gland atrophy and IM is a risk factor for late cancer. However, the authors did not check the state of the gastric mucosa before the eradication, so please revise the description.
Agree. Sentence of “the poor improvement of fundic gland atrophy” were revised to “the low remnant rate of fundic gland atrophy” according to reviewer’s comment.
- Please describe the reason for dividing the cases into the immediate group and the delayed group based on whether they were eradicated within 6 years or after more than 7 years.
Thank you for your comments. Indicated in Methods part, the incidence of pE-GC was bimodal after 7 years after Hp -ET. pE-GCs were abundant by 1 year after eradication, and remained the same thereafter, but decreased drastically after 7-8 years, and began to increase again after 9 years. Therefore, we divided the cases into two groups.
- Please explain the reason for excluding diffuse-type gastric cancer.
Thank you for your comments. Diffuse type gastric cancers were rare (n = 8) in this study and indicated very high rate of remnant rate of the fundic glands. So, we thought those to be bias and excluded from the analysis.
- The authors state the presence of IM as a mechanism that causes neutrophil infiltration of the gastric mucosa, but is there a link between neutrophil infiltration and the presence of IM in this study?
Thank you for pointing this out. We agree with this comment. Therefore, according to reviewer’s comment, we revised Table2 and Result part of manuscript. Intestinal metaplasia with neutrophil infiltration was added to Table2. Further, descriptions as to Table2 were revised as follows: Additionally, in both groups, most of IM cases were associated with neutrophil infiltration (79.4% in the immediate group,74.1% in the delayed group) (Table 2).
- As a limitation of this study, please include that the background tissue was analyzed using ESD resection specimens.
Thank you for your comments. According to reviewer’s comment, we added descriptions as follows: Second, this report used ESD specimens for the assessment of background mucosa. The findings of background mucosa might change depending on the site of the lesion.
- Please check whether the asterisks in Line 120 and 122 are necessary.
Thank you for your comments. According to reviewer’s comment, we deleted the asterisks.
Reviewer 2 Report
Comments and Suggestions for Authors
It is believed that Helicobacter pylori eradication (Hp-ET) can reduce the risk of gastric cancer but cannot completely prevent the occurrence of gastric cancer after eradication. In this retrospective study, the authors divided 100 qualified patients into two groups based on when gastric cancer was detected after Hp-ET: immediate (<6 years) and late (>6 years). By conducting analysis of existing clinical endoscopic and histological data on background and tumor mucosa from the two patient groups, the authors aimed to discover risk factors for developing posteradication gastric cancers (pE-GCs) after Hp-ET and provide information on surveillance. Unfortunately, no significant differences in mucosa were identified between the two groups. However, after comparison of other characteristic features of gastric cancer from the two groups, the authors suggested two risk factors for cancer development: 1) poor improvement in fundic glands atrophy and intestinal metaplasia; and 2) persistent mucosal inflammation in the mucosa. The work matches with the scope of the special issue focusing on Gastrointestinal Malignancies. The materials are organized well. Some minor language editing may improve readability of the paper.
Minor points:
1. In line 88, 120 cases may be changed to 100 cases to match with the legend of Figure 1.
2. The sentence from line 213 to line 215 needs to be revised.
3. In line 244, the word “with” seems not necessary.
Comments on the Quality of English LanguageSome minor language editing may improve readability of the paper.
Author Response
- In line 88, 120 cases may be changed to 100 cases to match with the legend of Figure 1.
Thank you for your comments. According to reviewer’s comment, we changed to 100 cases.
- The sentence from line 213 to line 215 needs to be revised.
Thank you for your comments. According to reviewer’s comment, we added descriptions as follows: In this study, pE-GCs with IM were significantly high not only in the immediate group but also in the delayed group and the delayed group had persistent neutrophil infiltration, which should be controlled by Hp -ET.
- In line 244, the word “with” seems not necessary.
Thank you for your comments. According to reviewer’s comment, we deleted the word “with”.
Reviewer 3 Report
Comments and Suggestions for Authors
This study by Abe et al. examined the endoscopic and histological features of late pE-GCs, comparing immediate and delayed groups. In the delayed group, the remnant fundic gland rate was 19.8% ± 15.6%, intestinal metaplasia (IM) was observed in 87.1%, and 90.3% of patients exhibited persistent neutrophil infiltration. These findings suggest that patients with poor improvement in fundic gland atrophy or IM and persistent inflammation are at higher risk for developing pE-GCs. The manuscript is well-written and presents an interesting study, with the limitations of the current work presented. I have only one comment: in the “Introduction” section, it is recommended to include a summary of the main results.
Author Response
This study by Abe et al. examined the endoscopic and histological features of late pE-GCs, comparing immediate and delayed groups. In the delayed group, the remnant fundic gland rate was 19.8% ± 15.6%, intestinal metaplasia (IM) was observed in 87.1%, and 90.3% of patients exhibited persistent neutrophil infiltration. These findings suggest that patients with poor improvement in fundic gland atrophy or IM and persistent inflammation are at higher risk for developing pE-GCs. The manuscript is well-written and presents an interesting study, with the limitations of the current work presented. I have only one comment: in the “Introduction” section, it is recommended to include a summary of the main results.
Thank you for your comments. According to reviewer’s comment, we added descriptions to Introduction part as follows: In this study, we focused on remnant rate of the fundic glands, intestinal metaplasia (IM), crypt enlargement, and neutrophil infiltration of the background mucosa.
Round 2
Reviewer 1 Report
Comments and Suggestions for Authors
Comments:
1. Additional investigations have shown differences between the immediate and the delayed groups in terms of history of endoscopic resection and multiple gastric cancers. Please search the previous reports and discuss thoroughly.
2. Line20-21,38-39,254-255: “the low remnant rate of fundic gland atrophy/IM” should be “the low remnant rate of fundic gland and IM”.
3. In Table 1, the 3/18 in the Multiple gastric cancers row should be 3/28.
4. Please correctly describe the p-value for the difference between the immediate and delayed groups in the history of endoscopic resection. Now, it is 0.000.
5. Since there are multiple gastric cancers, please revise the lesion characteristics in Table 1.
Author Response
- Additional investigations have shown differences between the immediate and the delayed groups in terms of history of endoscopic resection and multiple gastric cancers. Please search the previous reports and discuss thoroughly.
Thank you for pointing this out. We agree with this comment. We searched the previous reports in terms of endoscopic resection and multiple gastric cancers of increased or not in late pE-GCs. However, we found no any reports in these topics. Therefore, we mentioned in Discussion part as follows: In addition, the delayed group had more experience of ER and had more multiple gastric cancers at the same time than the immediate group in this study, although there was no previous report to support these results. From the above, we should be paid more attention to lesions in the middle third of the stomach, post ER cases and multiple lesions when monitoring pE-GCs during surveillance endoscopy.
- Line20-21,38-39,254-255: “the low remnant rate of fundic gland atrophy/IM” should be “the low remnant rate of fundic gland and IM”.
Thank you for your comment. According to reviewer’s comment, we revised terms like that.
- In Table 1, the 3/18 in the Multiple gastric cancers row should be 3/28.
Thank you for your comment. According to reviewer’s comment, we revised terms like that.
- Please correctly describe the p-value for the difference between the immediate and delayed groups in the history of endoscopic resection. Now, it is 0.000.
Thank you for your comment. According to reviewer’s comment, we revised terms the p-value for it to 0.0004.
- Since there are multiple gastric cancers, please revise the lesion characteristics in Table 1.
Thank you for pointing this out. We investigated the cases of multiple gastric cancers as one using the representative lesion. So the lesion characteristics were same and constant.Therefore,we added comments in Methods part as follows: In the cases of multiple gastric cancers, we investigated only one representative object (e.g. the largest lesion or the deepest lesion).